# Evaluation of EphB4 as Target for Image-Guided Surgery of Breast Cancer

**DOI:** 10.3390/ph13080172

**Published:** 2020-07-30

**Authors:** Cansu de Muijnck, Yoren van Gorkom, Maurice van Duijvenvoorde, Mina Eghtesadi, Geeske Dekker-Ensink, Shadhvi S. Bhairosingh, Alessandra Affinito, Peter J. K. Kuppen, Alexander L. Vahrmeijer, Cornelis F. M. Sier

**Affiliations:** 1Department of Surgery, Leiden University Medical Center, 2300 RC Leiden, The Netherlands; C.de_Muijnck@lumc.nl (C.d.M.); Y.van_Gorkom@lumc.nl (Y.v.G.); m.van_Duijvenvoorde@lumc.nl (M.v.D.); m_egh@hotmail.com (M.E.); N.G.Dekker-Ensink@lumc.nl (G.D.-E.); S.Bhairosingh@lumc.nl (S.S.B.); P.J.K.Kuppen@lumc.nl (P.J.K.K.); A.L.Vahrmeijer@lumc.nl (A.L.V.); 2Percuros BV, 2333 CL Leiden, The Netherlands; a.affinito@percuros.com; 3Department of Molecular Medicine and Medical Biotechnology, “Federico II” University of Naples, 80131 Naples, Italy

**Keywords:** breast carcinoma, EphB, tyrosine kinase receptor, targeted, image-guided surgery

## Abstract

Background: Targeted image-guided surgery is based on the detection of tumor cells after administration of a radio-active or fluorescent tracer. Hence, enhanced binding of a tracer to tumor tissue compared to healthy tissue is crucial. Various tumor antigens have been evaluated as possible targets for image-guided surgery of breast cancer, with mixed results. Methods: In this study we have evaluated tyrosine kinase receptor EphB4, a member from the Eph tyrosine kinase receptor family, as a possible target for image-guided surgery of breast cancers. Two independent tissue micro arrays, consisting of matched sets of tumor and normal breast tissue, were stained for EphB4 by immunohistochemistry. The intensity of staining and the percentage of stained cells were scored by two independent investigators. Results: Immunohistochemical staining for EphB4 shows that breast cancer cells display enhanced membranous expression compared to adjacent normal breast tissue. The enhanced tumor staining is not associated with clinical variables like age of the patient or stage or subtype of the tumor, including Her2-status. Conclusion: These data suggest that EphB4 is a promising candidate for targeted image-guided surgery of breast cancer, especially for Her2 negative cases.

## 1. Introduction

Breast cancer is the most commonly diagnosed cancer in women in Europe with an incidence of 523,000 new cases every year and is the third leading cause of death from cancer with 135,000 women every year [1]. Despite improved techniques, establishing tumor-negative margins remains the greatest challenge in breast cancer surgery for successful treatment and favorable patient outcome. With the increased appliance of breast conservative surgery, positive surgical margin rates of up to 40% have been reported in the literature, which leads to reoperation, less favorable cosmetic outcome, higher physical and psychological burden for the patients, and increased costs [2]. Even after re-excision, residual disease is detected at the surgical margins in up to 17% of the cases, constituting a major risk for local recurrence and metastases [3,4]. Methods of tumor boundary assessment vary from traditional tactile information, wire-guided localization, and analysis of frozen biopsies to advanced imaging technologies such as intraoperative positron emission tomography (PET) imaging and optical coherence tomography [2,5]. All these modalities have their limitations and none has yet proved to outperform the others. Targeted fluorescence image-guided surgery (FIGS) is a novel technique which aims to find a solution for this fundamental problem of oncological surgery.

FIGS relies on intraoperative visualization of tumor tissue by labeled tracers targeting tumor specific antigens. Although any overexpressed protein in tumor cells can be a potential target for FIGS, an ideal target should be abundantly present on the cell membrane of cancer cells in the majority of the tumors, but scarcely present in adjacent normal tissue [6,7]. Various tumor antigens have been documented in the literature as possible targets for FIGS, including a relatively high number of tyrosine kinase receptors like EGFR and cMet [6]. Regarding breast cancer, targeted FIGS seems to be particularly focused on Her2 and VEGF as tumor antigens, using established therapeutic monoclonal antibodies against these targets [8]. Considering that only a minority of breast cancers are Her2 positive and that VEGF is associated with angiogenic endothelial cells rather than with malignant cancer cells, there seems room for more specific targets for imaging of breast cancer.

EphB4 is a cell surface protein from the Ephrin tyrosine kinase receptor family. The members of the EphB subfamily are transmembrane proteins playing a role in various processes regarding tissue architecture and cellular growth, most prominently during the development of nervous and vascular systems. EphB4 and its ligand Ephrin B2 are involved in mammary morphogenesis in normal breast tissue [9]. EphB4 is also reported to be abundantly present in a variety of solid tumors, including colorectal cancer, prostate cancer, gastric cancer, and breast cancer, suggesting a common mechanism in carcinogenesis [10,11,12,13,14]. However, the role of EphB4 in tumorigenesis is way more complicated than in normal tissue, with tumor suppressing as well as tumor promoting effects [15]. As a cell surface protein overexpressed in a wide range of tumors, EphB4 seems to be a promising target for FIGS.

In this study we investigated the applicability of EphB4 as a target for breast cancer imaging. For this purpose we used immunohistochemical staining on two tissue microarrays (TMA) containing malignant as well as healthy breast tissues. TMA’s allow the evaluation of relatively large numbers of patients, which is necessary to establish the suitability of EphB4 for important subgroups of tumors, in particular low stage and Her2 negative tumors.

## 2. Results

The patient and tumor characteristics from a test and validation TMA of breast cancer tissue are presented in Table 1. The characteristics and number of patients were comparable between both cohorts, except for the mean age of the patients, which was slightly but significantly higher in the test cohort (58 ± 13 years versus 57 ± 13, *p* = 0.027), and there were less early stage patients present in the test cohort (*p* = 0.000). None of the patients in the test cohort received neoadjuvant therapy, whereas in the more recent validation cohort less than 10 percent of the patients did.

Irrespective of the type of antibody used, immunohistochemical staining for EphB4 was primarily present in tumor cells throughout the tumor tissue cores. EphB4 staining was prominently observed on the membrane and in the cytoplasm of the cells and occasionally in the nucleus (Figure 1). In general, tumor tissue showed more intense staining than normal tissue. But comparison of T/N pairs showed the presence of cases where normal tissue expression was equal or higher than in the corresponding tumor. Figure 1A shows a typical example of a patient with enhanced staining of EphB4 in tumor epithelial cells and low staining for corresponding normal tissue. Figure 1B shows a particular case with more staining in normal tissue than corresponding tumor tissue. For scoring of the staining’s, the mouse monoclonal antibody sections were used for both TMAs. The staining of all tissues was scored by two independent examiners, showing a strong correlation (kappa = 0.824, *p* = 0.001). For some cases interobserver agreement was obtained by re-evaluation of the respective sections, including a third opinion (CS).

From the 662 patients in the test cohort, the staining results of 400 tumors and 56 normal tissue samples were suitable for comparative evaluation. In the 400 tumors, 397 (99.2%) were positive for EphB4 and there were only 3 cases (0.8%) with a score of 0. In the 39 paired samples (T/N) the mean score for EphB4 staining was 4.8 ± 1.7 in tumor tissue versus 4.0 ± 1.5 in normal tissue. A paired test showed that the staining was higher for tumor than for normal tissue, but the difference was not statistically significant (*p* = 0.055), probably due to the low number. Characteristics of the subgroup of patients with complete T/N sets which were used for paired analysis are reported in Table 2.

The validation TMA contained 667 patients of which 590 tumors and 71 normal tissue were usable for further analysis, resulting in total 61 complete T/N sets. A paired *t*-test confirmed the difference in EphB4 between tumor and normal tissue, with a *p*-value of 0.03. For image-guided surgery applications, the difference in expression of a target protein between tumor and normal tissue is of paramount importance. Figure 2 indicates in a graphical representation the immunohistochemical scores of tumor and normal pairs for individual patients in both cohorts. From the 100 T/N sets in both cohorts, EphB4-based imaging would have been appropriate for 60 patients according to their immunohistochemistry (IHC) T/N profile, as indicated by the green lines.

The mean score of EphB4 staining did not vary significantly between different tumor stages in the test cohort and validation cohort. No significant difference was observed between EphB4 staining and different age groups or tumor type, *p* values being 0.625 and 0.859 respectively in test cohort. More importantly, there was no difference between the mean score of Her2 negative tumors and tumors with Her2 overexpression (*p* = 0.552). A summary of EphB4 staining for tumor and normal tissues per tumor stage for the whole test cohort and the subgroup of patients with Her2 negative tumors are presented in Figure 3. In the validation cohort, neoadjuvant therapy was not associated with decreased EphB4 staining, *p* = 0.129.

To substantiate further the applicability of EphB4 as target for breast cancer imaging, we have investigated the binding of the EphB4 polyclonal antibody used for IHC, which detects extracellular, domains, for flow cytometry on a panel of breast cancer cell lines selected for respectively low (MCF-10A), intermediate (MCF-7 and MDA-MB-231) and high (HCC1954) expression of Her2. Figure 4 indicates that all breast cancer cell lines were positive for EphB4, irrespective of the presence of Her2.

## 3. Discussion

Most previous investigations of breast cancer imaging have been focused on Her2 over-expression as target. Caused by gene-multiplications, the mutated breast tumor cells present with a disproportional number of Her2 copies on their membrane in comparison with normal breast cells. (Pre)clinical studies with Her2 targeting antibodies like trastuzumab and pertuzumab have substantiated the principle of Her2-targeted imaging [16]. Unfortunately, Her2 overexpression is found in less than 30% of the patients. Furthermore, Her2 status shows heterogeneity between tumor, lymph node, and organ metastasis and up to 30% of patients are reported to switch Her2 receptor status following a neoadjuvant therapy with anti-Her2 directed antibody trastuzumab [17].

Therefore, we have focused on EphB4 as an alternative target, especially for Her-2 negative breast cancer patients. Members of the Ephrin receptor family have already been indicated for their role in carcinogenesis in various types of tumors [10,11,12,13,14,18,19,20]. As a result, EphA2 is highly ranked in the NIH’s list for potential cell surface antigen targets for cancer treatment [21]. In a comparative study we have shown that EphA2 and EphB4 indeed seem promising targets for image-guided surgery for colon cancer, but with EphB4 having more consistent results. In the present study we evaluated EphB4 in breast cancer specimen using two TMAs consisting of in total 1329 patients with 100 complete T/N tissue pairs. To our knowledge, this is the largest cohort studied with immunohistochemistry so far for EphB4 expression in breast cancer.

Presence on the surface of tumor cells is the main prerequisite for targeted cancer imaging. As a transmembrane tyrosine kinase receptor, EphB4 is expected to be mainly present on the surface of cells. We noticed occasional staining of EphB4 in the cytoplasm and nucleus, comparable with was previously described in breast cancer cell lines [22]. However, the vast majority of the expression was located at the membrane of malignant epithelial cells, generally throughout the whole tumor section, qualifying EphB4 as an appropriate protein for targeting.

The next important feature for a tumor target to be applicable for FIGS is the expected T/N ratio in vivo, also known as tissue background ratio (TBR). TBR is defined as the ratio of mean fluorescence signal of the tumor to the mean signal of surrounding healthy tissue. A favorable TBR is seen as a cornerstone for the clinical translation of a fluorescence probe. To be able to recognize malignancy during an operation, the expression in normal tissue should preferably be absent. But recent (pre)clinical studies indicate that a ratio of at least 2 in favor of the tumor might suffice for other tumor targets like EpCAM [23]. Presence in cytoplasm will not interfere with targeted imaging; it could even be an advantage if EphB4 would be internalized after being targeted by a tracer. Our data on 100 T/N sets indicate that for the majority of breast cancers EphB4 would be an appropriate target. A recent overview of molecular tracers for FIGS in clinical studies indicates relatively few targets being under investigation for breast cancer, i.e., folate receptor, VEGF, chlorotoxin binding proteins, HSP-90, integrins, cathepsins, and MMPs [24]. Although most of these targets are supported by immunohistochemical studies for various types of cancer, very few if any of these targets are evaluated in T/N comparisons, like shown here for EphB4. This makes impartial comparison of targets difficult. Comparative immunohistochemical studies, especially focused on imaging application, are able to support target selection for various types of cancer [25,26,27,28].

It is important for a FIGS target to be over-expressed in all stages of the disease. While there are some studies in the literature reporting a correlation between tumor stage and expression of EphB4, we observed no such association [20,29]; in our relatively large cohort EphB4 staining did not vary significantly between different tumor stages. Our findings support EphB4 as a good target for FIGS, even for early stage breast cancer patients. Unfortunately our cohort did not contain lymph nodes, but other studies report EphB4 to be (over)expressed in positive lymph nodes of various cancers [30]. If further substantiated that an EphB4 based near infrared fluorescence (NIRF) tracer could be used to detect positivity in sentinel lymph nodes, next to, or instead of the non-targeted near-infrared fluorescent agent indocyanine green (ICG). Sentinel lymph node mapping (SLM) with ICG has been shown to have comparable detection rates with standard-of-care radioactive tracers and blue dye [31,32,33]. Staining of metastatic cells in lymph nodes would substantially enhance the possibilities of fluorescent imaging and circumvent the limitations of ICG, such as poor aqueous stability, concentration-dependent aggregation, short half-life in the circulation, and lack of target specificity.

Like most investigations, this study has several limitations, like the already mentioned absence of positive lymph nodes. Furthermore, semi-quantification of immunohistochemical staining is subjective and rapidly developing imaging systems and software will soon replace and outperform visual scoring by experts. The use TMA sections consisting of 3 cores per tumor makes the evaluation of intertumoral heterogeneity challenging in comparison with whole tumor sections. Still, TMAs offer a big advantage with respect to obtaining data from fairly large cohorts, like in this study. Furthermore, our cohort(s) were collected from 1985 to 2009, which means that only few patients were pre-operatively treated with neoadjuvant therapy, the present standard of care. Because anti-Her2 antibodies are often integrated in in adjuvant therapy schedules, alternative imaging targets like EphB4 could be advantageous.

Once EphB4 is accepted as a serious candidate for FIGS a targeting vehicle has to be developed. Various agents such as antibodies or peptides are being investigated to target Eph receptors, also for image guided purposes [34,35,36]. Peptides have the advantages of low toxicity, highly efficient tissue penetration, and generally lower costs comparing to biologicals [35]. On the other hand, antibodies offer high tumor specificity and affinity and could be modified, like recently developed bi- and tri-specific versions antibodies to enhance specificity depending on the purpose [37,38]. Liu et al. have visualized in vivo tumors successfully with a series of anti EphB4 antibodies for targeted imaging with PET in mice xenografted with human HT-29 colon cancer and MDA-MB-231 breast cancer cells (38). Others have developed near infrared florescence probes with EphB4 antibodies for targeted tumor imaging (34). Another subclass of probes which are extensively investigated for PET imaging are small molecules. Small molecules have the ability to cross human body’s natural barriers such as the blood–brain barrier, more easily. This feature makes them more eligible as a probe for imaging/therapy of certain tumors like central nervous system malignances. Alternatively, in a recent publication, Affinito et al. introduced a RNA aptamer targeting EphA2 receptors at human glioblastoma cells for therapeutic purpose-s [39]. In another recent study, Pretze et al. reported two novel xanthine derivatives as potential tracers for imaging of EphA2 and EphB4 overexpressing human A375 melanoma cells [40]. Although the suitability of those probes still needs to be evaluated in clinical trials, these studies globally underscore the potency of EphB4 as a target for tumor imaging

## 4. Materials and Methods

### 4.1. Patients and Tumors

Tissue micro arrays (TMAs) derived from two consequent cohorts of patients with breast cancer patients were used. The TMAs, both containing sets of normal and cancer tissue, were sectioned and stained for EphB4, the first cohort being the test set and the second the validation set. The test set included all patients diagnosed and operated for breast cancer in the Leiden University Medical Center (LUMC) between 1985 and 1996 (*n* = 662) [41] and the validation set included the patients from 1996 until 2009 (*n* = 667). There were no overlapping patients in the two cohorts. The local ethics review board (Medische-Ethische Toetsingscommissie Leiden Den Haag Delft (METC-LDD)) approved the study protocol and research was conducted according to the Code Goed Gebruik (Human Tissue and Medical Research: Code of conduct for responsible use (2011)) and to the Code Goed Gedrag (Code of Conduct for Medical Research (2004)). Both codes are prescribed by the Dutch Federation of Medical Scientific Societies. Informed consent was not needed for this study, because samples and data were non-identifiable and used in accordance with the 1964 Helsinki declaration.

### 4.2. Tissue Micro Array

Formalin-fixed paraffin-embedded (FFPE) tissue blocks of primary tumors and their respective normal tissues were collected from the LUMC Pathology department. Sections were cut for hematoxylin–eosin staining and pathologically representative tumor regions were used for the preparation of TMA blocks. Three tissue cores were punched from each donor block (respectively 0.6 mm in test cohort and 1 mm for the validation set) from tumor areas and transferred into a recipient paraffin block with a TMA master (3DHistech, Budapest, Hungary). The cores were taken from three different locations across the tumor tissue. For approximately one third of the patients (277 for the test cohort and 135 for the validation cohort) three cores were available from normal breast tissue distal to the tumor.

### 4.3. Immunohistochemistry

IHC staining was performed on 4 m sections cut from each TMA receiver block. TMA sections were deparaffinized in xylene and rehydrated through a series of graded alcohol to PBS. Endogenous peroxidase was blocked for 20 min in 0.3% hydrogen peroxide in water. The sections were treated for antigen retrieval in Envision Flex Target Retrieval solution low, pH6 (DAKO Cytomation, Glostrup, Denmark) for 10 min at 95 °C (DAKO PT Link, Glostrup, Denmark). Sections were incubated overnight with primary antibodies. Antibodies used for EphB4 staining were monoclonal mouse IgG1 (37-1800, Life Technologies/Thermo Fisher Scientific, Waltham, MA, USA) and polyclonal goat (AF3038, R&D Systems). The optimal antibody dilutions for staining of breast cancer tissue was determined independently for both antibodies. After 30 min of incubation with DAKO anti-mouse envision or rabbit anti-goat antibodies with HRP (respectively K4001 and K4003, DAKO Cytomation), the sections were visualized using a diaminobenzidine solution (DAB+; DAKO Cytomation), resulting in brown color. The sections were counter stained with hematoxylin, dehydrated, and mounted with pertex (Histolab, Gothenburg, Sweden). The entire slides were scanned with a Philips Ultra Fast Scanner 1.6 RA (Philips, Eindhoven, The Netherlands) for further analysis.

### 4.4. Scoring Method

Each TMA set was semi-quantitively scored by two independent examiners. A TMA core was only assessed if more than 50% of the core was occupied by tissue. The intensity of the staining was evaluated as 0 when there was no staining, 1 for weak staining, 2 for moderate staining, and 3 for intense staining. The percentage of cells stained was scored 0 for 0%; 1 when <25% of the cells were stained; 2 for 25–50% staining; 3 for 50–90% staining, and 4 when more than 90% of the cells were stained. Only tumors with 2 or more scores were used for data analysis. The median of the intensity score and the median of the percentage score were added up and formed a final score (0–7) for each tissue. For comparative statistics only complete sets of tumor-normal tissue were used.

### 4.5. Flow Cytometry

Breast cancer cell lines MCF-10A, MCF-7, MDA-MB-231 and HCC1954, as well as Jurkat (leukemic T-cell lymphoblast) as negative controls were grown in RPMI or DMEM (Gibco, Life Technologies, Carlsbad, CA, USA) as appropriate, with 10% fetal calf serum and 100 IU/mL penicillin/streptomycin (Gibco) at 37 °C in a humidified incubator with 5% CO_2_. The presence of EphB4 on the membranes of these cells was determined by flow cytometry. Cells were cultured until 90% confluence and detached with trypsin/EDTA. Viability of the cells was evaluated with trypan blue. The cells were incubated with 4 μg/mL polyclonal goat antibodies AF3038 against EphB4 for 30 min on ice, washed with ice cold phosphate buffered saline pH7.5 (PBS), and incubated with anti-goat secondary antibody conjugated with FITC (A11078, Life Technologies). The cells were then centrifuged, washed, and suspended in PBS containing propidium iodide to exclude dead cells, and consequently analyzed in a BD LSRII flow cytometer (BD Biosciences, San Jose, CA, USA) using FlowJo software (Tree Star Inc., Ashland, OR, USA).

### 4.6. Statistical Analysis

Statistical analyses were conducted using SPSS statistical software (version 20.0 for Windows, SPSS Inc., Chicago, IL, USA) and R Software: R2.12.1 (http://www.r-project.org/). Scores are presented as mean ± standard deviation. The difference between tumor and normal tissue was calculated using Student’s paired *t*-test for paired samples and Independent Samples *t*-test for unpaired data in Figure 3. For the difference in tumor stage and age groups ANOVA tests were used. Independent Samples *t*-test was used for the assessment of mean staining for Her2 positive versus negative tumors and tumors with neoadjuvant therapy versus without neoadjuvant therapy. Inter-rater reliability was assessed by calculating kappa values. All statistical tests were conducted two-sided, and p-values of 0.05 or less were considered significant. 

## 5. Conclusions

Based on immunohistochemical results from two relatively large cohorts of in total 1329 patients EphB4 shows potential as a target for image-guided breast cancer surgery. Overexpression of EphB4 seems to be applicable for imaging purposes irrespective of the stage of the tumor. Importantly, because EphB4 overexpression is not associated to Her2 status, further preclinical investigation is needed to establish whether this target could be applied for imaging of the majority patients with Her2 negative breast cancer.

## Figures and Tables

**Figure 1 pharmaceuticals-13-00172-f001:**
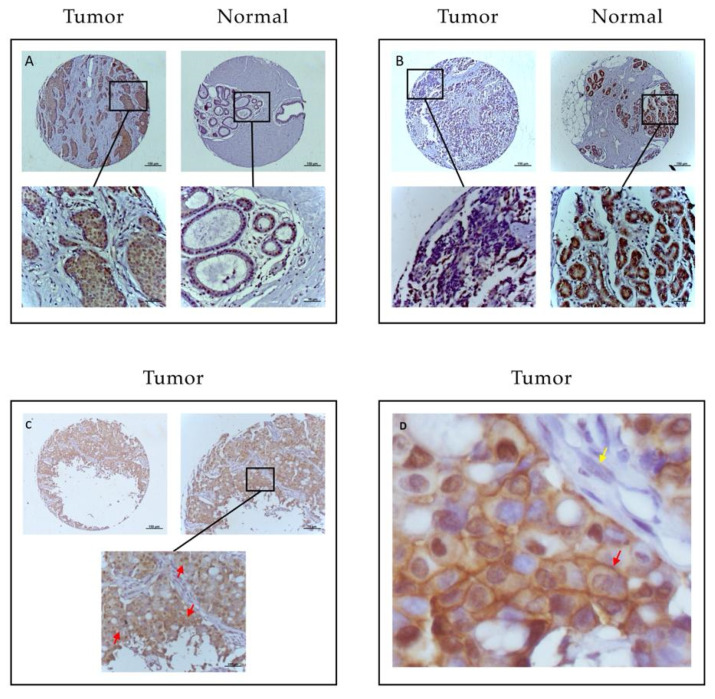
Examples of staining pattern for EphB4 in tumor and normal tissue sets from patients with breast cancer. (**A**) shows the common staining pattern with higher expression of EphB4 in tumor tissue comparing to corresponding normal tissue. (**B**) shows an aberrant pattern with higher expression of EphB4 in normal tissue than staining in tumor tissue. The scale bars represent 100 micrometers at pictures above. Pictures below show 40× enlargements of the sections in (**A**–**C**). (**D**) shows 400× enlargement of a tumor section from panel (**C**) to emphasize membranous staining of EphB4. Red arrows indicate membrane staining, yellow arrow absence of EphB4 in stromal cells.

**Figure 2 pharmaceuticals-13-00172-f002:**
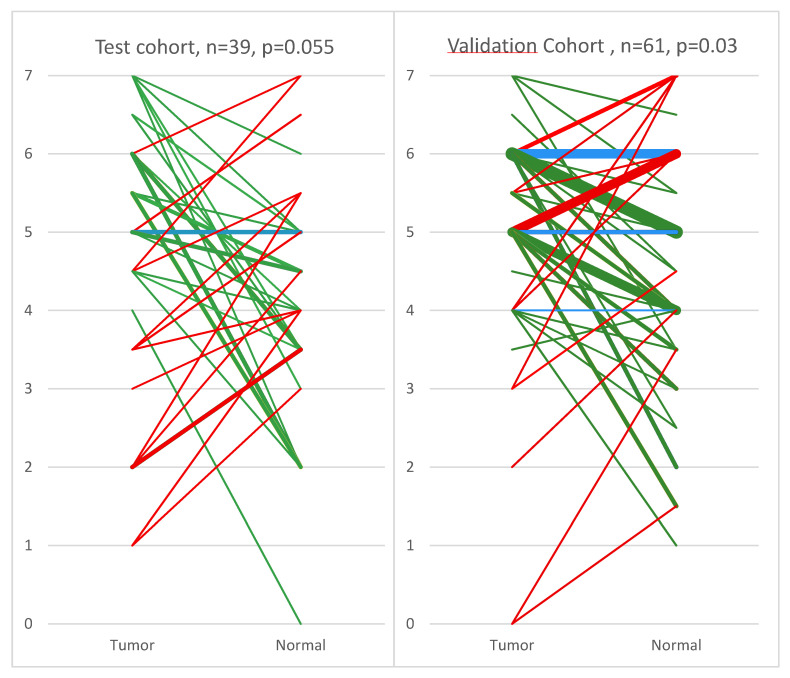
Scoring diagram of breast cancer and adjacent normal tissue pairs of the test cohort (**A**) and validation cohort (**B**) stained for EphB4. Green = Tumor is higher score than normal; Red = Normal tissue is higher score than tumor; Blue = Tumor and normal tissue share the same score. The thickness of the line is proportional to the number of sets with identical scores. The numbers 0–7 on the *y*-axis represent the IHC staining scoring values. The thickest line (in panel B, green line 6 to 5) represents 7 pairs and the thinnest line (in both panels) represents 1 pair.

**Figure 3 pharmaceuticals-13-00172-f003:**
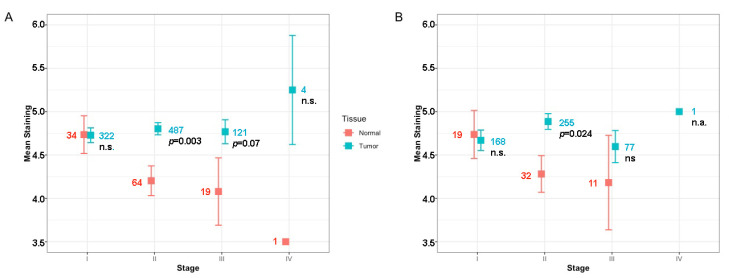
Comparison of mean EphB4 staining score per tumor stage in breast cancer for tumor and corresponding normal breast tissue from 934 patients (**A**). The same comparison for the 501 patients with Her2 negative tumors selected from the total cohort (**B**). Blue and red colors indicate tumor and normal tissue. Error bars represent mean ± SD. Numbers of tissues per stage are indicated with similar color code. n.s., not significant, n.a., not applicable.

**Figure 4 pharmaceuticals-13-00172-f004:**
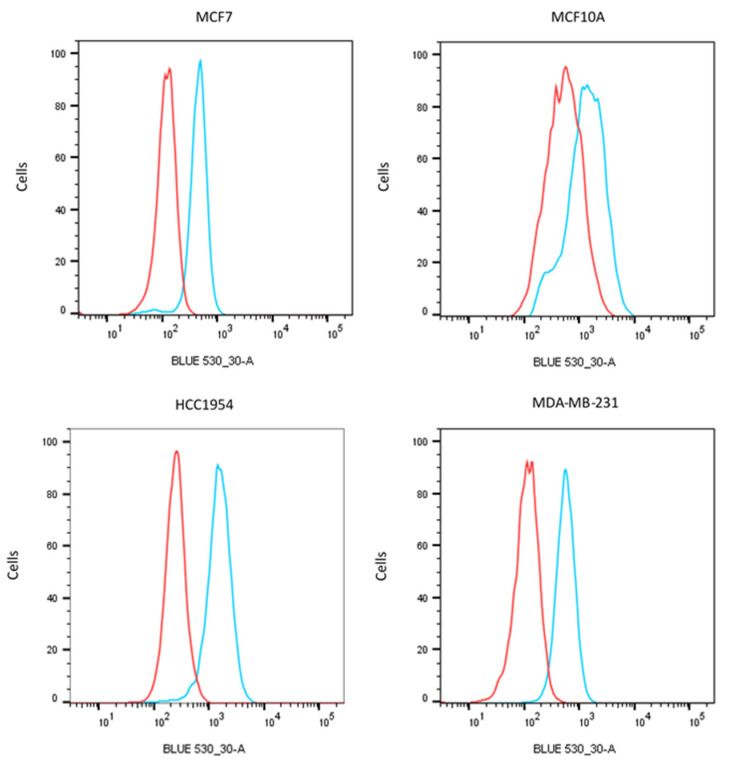
Presence of EphB4 on breast cancer cell lines MCF-10A, MCF7, MDA-MB-231, and HCC1954 indicated in blue versus control in red.

**Table 1 pharmaceuticals-13-00172-t001:** Characteristics of breast cancer patients in the test and validation cohort.

Characteristic	Test Cohort		Validation Cohort	
	*n* (662)	%	*n* (667)	%
Age in years				
<45	124	18.7	134	20.1
45–55	163	24.6	214	32.1
55–65	149	22.5	148	22.2
>65	222	33.5	171	25.6
Missing	4	0.6	0	0
Stage				
I	181	27.3	252	3.8
II	352	53.2	317	47.5
III	95	14.4	58	8.7
IV	0	0	4	0.6
Unknown	34	5.1	36	5.4
Histological type				
Ductal	588	88.8	536	80.4
Lobular	62	9.4	66	9.9
Other	0	0	65	9.7
Missing	12	1.8	0	0
Differentiation				
Good	109	16.5	108	16.2
Moderate	322	48.6	275	41.2
Poor	221	33.4	209	31.3
Unknown	10	1.5	75	11.2
Estrogen receptor				
Positive	368	55.6	456	68.4
Negative	272	41.1	140	21.0
Missing	22	3.3	71	10.6
Progesterone receptor				
Positive	327	49.4	317	47.5
Negative	299	45.2	260	39.0
Missing	36	5.4	90	13.5
Her2/Neu status				
Positive	52	7.9	84	12.6
Negative	495	74.8	247	37.0
Missing	115	17.4	336	50.4
Neoadjuvant therapy				
CT	0	0	32	4.8
HT	0	0	22	3.3
CT + HT	0	0	1	0.1
None	658	99.4	610	91.5
Missing	4	0.6	2	0.2

*n*, number of patients; CT, chemotherapy; HT hormonal therapy; CT + HT chemotherapy and hormonal therapy.

**Table 2 pharmaceuticals-13-00172-t002:** Characteristics of subgroup of 100 breast cancer patients in test and validation cohort.

Characteristic	Test Cohort		Validation Cohort	
	*n* (39)	%	*n* (61)	%
Age				
<45	14	35.9	21	34.4
45–55	9	23.1	20	32.8
55–65	5	12.8	12	19.7
>65	11	28.2	8	13.1
Missing	0	0	0	0
Stage (%)				
I	6	15.4	20	32.8
II	24	61.5	28	45.9
III	8	20.5	6	9.8
IV	0	0	1	1.6
Unknown	1	2.6	6	9.8
Histological type				
Ductal	32	82.1	53	86.9
Lobular	7	17.9	2	3.3
Other	0	0	6	9.8
Missing	0	0	0	0
Differentiation				
Good	5	12.8	10	16.4
Moderate	16	41.0	24	39.3
Poor	18	46.2	23	37.7
Unknown	0	0	4	6.6
ER				
Positive	21	53.8	41	67.2
Negative	18	46.2	14	23.0
Missing	0	0	6	9.8
PG				
Positive	21	53.8	29	47.5
Negative	18	46.2	21	34.4
Missing	0	0	11	18
HER2				
Positive	3	7.7	10	16.4
Negative	32	82.1	17	27.9
Missing	4	10.3	34	55.7
Neoadj therapy				
CT	0	0	3	4.9
HT	0	0	2	3.3
CT + HT	0	0	0	0
None	39	100	56	91.8
Missing	0	0	0	0.2

*n*, number of patients; CT, chemotherapy; HT hormonal therapy; CT + HT chemotherapy and hormonal therapy.

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
