# Peer review of "Evaluation of EphB4 as Target for Image-Guided Surgery of Breast Cancer"

_pharmaceuticals, 2020, doi:10.3390/ph13080172_

Round 1

Reviewer 1 Report

Overall “ Evaluation of EphB4 as target for image-guided 2 surgery of breast cancer” is an interesting and well-written paper.

MAJOR

The main concern is about the analysis and the comments arising from Figure 2 and table1. In fact, characteristics of breast cancer patients are reported in table 1 (662+667patients), whereas figure 2 report a subset of 39+61 patients selected within the previous patients in table 1. It would be useful to add a new table reporting the characteristics of breast cancer patients of figure 2 and revise comments consequently.

MINOR

Page 3, table 1. Stage 1 validation cohort: there is a typo in the percentage

Figure 2 The thickness of the line is proportional to the number of sets with identical scores, however it is not possible to understand how many samples represent a bold line. I suggest to split lines avoiding overlapping, add a table or any other convenient solution.

Line 190: this statement is based on an old report (2009). I wonder if there is any update

Line 204: this paragraph deals with T/N ratio however T/N ratio of EPhB4 was not reported in the results

Line 239-246: I would add some comments on small molecules in addition to peptides and antibodies.

Overall the paper: Ephb4 should be fixed as EphB4

Author Response

Reviewer 1

Reviewer 1:

Major

The main concern is about the analysis and the comments arising from Figure 2 and table1. In fact, characteristics of breast cancer patients are reported in table 1 (662+667patients), whereas figure 2 report a subset of 39+61 patients selected within the previous patients in table 1. It would be useful to add a new table reporting the characteristics of breast cancer patients of figure 2 and revise comments consequently.

  • We thank the reviewer for the suggestion and partly agree. Table 1 shows clearly that both cohorts are representative for breast cancer patients. In combination with figure 3 these data suggest that EphB4 is indeed over-expressed in the majority of tumors in a large, representative cohort. Figure 2 on the other hand shows essential information for cancer imaging, the T/N ratio for EphB4, which has never been presented before for breast cancer. Unfortunately, this analysis had to been done on a selection of the tumors, due to the fact that normal tissue is generally not taken into account for pathological examination and therefore not stored. Despite the relatively high number of 100 sets of T/N tissues, this sub-cohort of patients cannot be considered representative. Nevertheless, we have added a new table with the characteristics of the subset of 100 patients at line 137 and revised the comments accordingly at line 125 , as suggested by the reviewer. Because of the insertion of a new table , Figure 2 is moved to line 94 to assure a coherent layout.

Minor

Page 3, table 1. Stage 1 validation cohort: there is a typo in the percentage

  • Two comma’s are converted to dots in Table 1 at following parts: Stage I validation cohort percentage and Unknown stage test cohort percentage.

Figure 2 The thickness of the line is proportional to the number of sets with identical scores, however it is not possible to understand how many samples represent a bold line. I suggest to split lines avoiding overlapping, add a table or any other convenient solution.

  • We understand the concerns of the reviewer and partly agree. However, because the data in Figure 2 are discussed extensively in the results section, the purpose of the figure is to give a simple, graphical overview, which could never be achieved in a table.     The lines in Figure 2 represent in total 100 tumor/normal tissue pairs, with more than 50 possible scoring pairs. Separating the lines would make the figure simply too complicated. To put the thickness of the lines more into perspective, we have now indicated in the legend of figure 2 that the thickest line (in panel B) represents 7 pairs and the thinnest line (in both panels) represents 1 pair(Line 184). Tone difference in the colors are corrected. In the original figure, lines in Panel A were one time bolder than Panel B, that is undone. The thinnest lines at both panels are now identically thin, representing 1 pair.

Line 190/224: this statement is based on an old report (2009). I wonder if there is any update

  • ‘’Prioritization of cancer antigens’’ was a onetime pilot project of National Cancer Institute run by translational research work group. There are no following papers regarding the current status of the list.

Line 204/236: this paragraph deals with T/N ratio however T/N ratio of EPhB4 was not reported in the results

  • We appreciate the remark of the reviewer. The key word in the first sentence of this paragraph is ‘’expected’’ before T/N ratio. T/N ratio, also referred to as tissue to background ratio (TBR) is of essential importance for the clinical translation of a fluorescence probe. Because we are talking about an ‘’expected’ value in future preclinical/animal studies, it is obviously not reported in the results section of this study. To make our statement clearer, we revised the first sentence of the paragraph and added two more sentences at the line 249 defining TBR and emphasizing its value in clinical translation.

Line 239-246: I would add some comments on small molecules in addition to peptides and antibodies.

  • Two new sentences are added to line 296 about small molecules.

Overall the paper: Ephb4 should be fixed as EphB4

  • Revised in following lines: 63, 78, 84, 192, 198, 202, 205, 241, 265, 266, 296, 378, 379
    •  

Reviewer 2 Report

In the present manuscript de Muijnck et al investigate the usability of EphB4 as a possible target for fluorescence image-guided surgery (FIGS). Using an unexampled tissue microarray collection of breast cancer and related normal tissue samples the authors show the specificity of EphB4 toward cancerous, but not healthy cells. The results were validated on a different sample set and also on widely used cells lines with flow cytometry.

The manuscript is nicely written and some of the results are convincing, however there are some points which should be strengthen slightly better or organized differently.

1. On Figure 1 the authors claim that panel C proves EphB4 staining is mainly localized in the cell membrane of the cells, however the results shown here are not convincing. Using the same cell line panel from Figure 4 (MCF-10A, MCF7, MDA-MB-231 and HCC1954) a series of immunofluorescent staining combined with confocal imaging would be really helpful to prove the (mostly) membrane specificity of the EphB4 signal. Due to membrane-selectivity could be an important aspect for later FIGS probes, it’s an important issue which have to be addressed.

2. On Figure 3 tumoral EphB3 scores were compared to healthy tissue in regard to tumor stages. Stage IV have only a few (if any) samples to compare which it makes this comparison questionable. Is there a reason to show Stage IV data on these panels? Also, in the other stages, for example panel B Stage III, the difference is missing. Please calculate the significance of the differences by stages and indicate it on the figure.

3. In the Materials and Methods section under the 4.4. Scoring Method part, the authors state that “Only tumors with 2 or more scores were used for data analysis.” Why? Doesn’t it distort the whole analysis?

Minor issues:

Page 5 line 158: The sentence “More importantly.” standing alone. I think it’s a part of the following sentence, no?

Page 5 line 161: figure 3. Please always cite the Figures with capital F throughout the whole manuscript.

If the authors will address these issues in the revised version of the manuscript, I will recommend this work to be published in MDPI’s Pharmaceuticals.

Author Response

Reviewer 2

Major

On Figure 1 the authors claim that panel C proves EphB4 staining is mainly localized in the cell membrane of the cells, however the results shown here are not convincing. Using the same cell line panel from Figure 4 (MCF-10A, MCF7, MDA-MB-231 and HCC1954) a series of immunofluorescent staining combined with confocal imaging would be really helpful to prove the (mostly) membrane specificity of the EphB4 signal. Due to membrane-selectivity could be an important aspect for later FIGS probes, it’s an important issue which have to be addressed.

  • We agree completely with the reviewer that membrane selectivity is essential for FIGS probes. Tyrosine kinase receptors, like EphB4, are therefore amongst the most preferred targets. To show the abundant presence of EphB4 on epithelial cells, we included now a 10 times more magnification in Figure 1 as Panel D. The flow cytometry experiments were added to confirm the presence of EphB4 on the cell membranes of breast cancer cells. The antibody used for flow cytometry experiments detects extracellular domains of the receptor, as a result any signal that is presented in the charts is due to the staining of extracellular domains of membrane bound receptors. We believe that the IHC stainings in combination with the flow cytometry results substantiate membrane staining sufficiently. Legend of Figure 2 is adjusted based on the changes.

On Figure 3 tumoral EphB4 scores were compared to healthy tissue in regard to tumor stages. Stage IV have only a few (if any) samples to compare which it makes this comparison questionable. Is there a reason to show Stage IV data on these panels? Also, in the other stages, for example panel B Stage III, the difference is missing. Please calculate the significance of the differences by stages and indicate it on the figure.

  • We understand the concerns of the reviewer. However our primary goal with this figure was not per se to show a significant difference in staining of tumor and normal tissue per tumor stage, but to indicate the general enhancement of EphB4 in tumors, irrespective from the stage. This is why we didn’t exclude patients based on other characteristics. Nevertheless, we have indicated the p-values per stage, as suggested. Among the 100 patients which were eligible for paired analyses, there was unfortunately only 1 patient with stage IV disease. Additional information about the statistics of this figure is inserted in ‘’Materials and Methods’’ section at line 370.

In the Materials and Methods section under the 4.4. Scoring Method part, the authors state that “Only tumors with 2 or more scores were used for data analysis.” Why? Doesn’t it distort the whole analysis?

  • We agree with the reviewer that undefined selection of samples would bias the results. To obtain high enough number of samples we have used TMA’s rather than whole tissue sections for this investigation. These TMA’s were not specifically designed for the purpose of target evaluation, meaning that the 3 cores per tissue sample were taken from representative areas. The ideal location for target evaluation would be from the border of the tumor, because presence of the target in the core of the tumor will not contribute much to targeting with the purpose of imaging. Although the core size of 0.6 mm in our TMA’s was relatively large, we sincerely believe that tumors (or normal tissue) from which only one core was available should be eliminated, as not being representative enough for the purpose of target evaluation. This ‘selection’ criteria was defined before the actual analysis.

Minor

Page 5 line 158/190: The sentence “More importantly.” standing alone. I think it’s a part of the following sentence, no?

  • The dot is removed and “More importantly’’ is connected with the following sentence.

Page 5 line 161: figure 3. Please always cite the Figures with capital F throughout the whole manuscript.

  • Lower case “f” is converted to capital ‘’F’’s at line 204.

Correction not suggested by reviewers:

Line 369 - Not relevant heading ‘ Supplementary Materials’ has been removed

Round 2

Reviewer 1 Report

Thank you for the revision.

Now it is fine for me

Reviewer 2 Report

As a thought-provoking first publication for this topic the manuscript is fine. Hopefully the authors will pursue EphB4 as a possible target for FIGS further and then we can learn more scientific details and read more interesting papers about this possibility later!

Thanks for the authors for the revision!